# Ensemble data assimilation to diagnose AI-based weather prediction

model: A case with ClimaX version 0.3.1

- 4 Shunji Kotsuki<sup>1,2,3</sup>, Kenta Shiraishi<sup>4</sup>, and Atsushi Okazaki<sup>1,2</sup>
- <sup>1</sup>Institute for Advanced Academic Research, Chiba University, Chiba, Japan
  - <sup>2</sup>Center for Environmental Remote Sensing, Chiba University, Chiba, Japan
- <sup>3</sup>Research Institute of Disaster Medicine, Chiba University, Chiba, Japan
- 8 <sup>4</sup>Graduate School of Science and Engineering, Chiba University, Chiba, Japan

Correspondence to: Shunji Kotsuki (shunji.kotsuki@chiba-u.jp)

#### Abstract.

Artificial intelligence (AI)-based weather prediction research is growing rapidly and has shown to be competitive with the advanced dynamic numerical weather prediction models. However, research combining AI-based weather prediction models with data assimilation remains limited partially because long-term sequential data assimilation cycles are required to evaluate data assimilation systems. This study proposes using ensemble data assimilation for diagnosing AI-based weather prediction models, and marked the first successful implementation of ensemble Kalman filter with AI-based weather prediction models. Our experiments with an AI-based model ClimaX demonstrated that the ensemble data assimilation cycled stably for the AI-based weather prediction model using covariance inflation and localization techniques within the ensemble Kalman filter. While ClimaX showed some limitations in capturing flow-dependent error covariance compared to dynamical models, the AI-based ensemble forecasts provided reasonable and beneficial error covariance in sparsely observed regions. In addition, ensemble data assimilation revealed that error growth based on ensemble ClimaX predictions was weaker than that of dynamical NWP models, leading to higher inflation factors. A series of experiments demonstrated that ensemble data assimilation can be used to diagnose properties of AI weather prediction models such as physical consistency and accurate error growth representation.

#### 1 Introduction

The intensification of weather-induced disasters due to climate change is becoming increasingly severe worldwide (e.g., Jonkman et al. 2024). In a recent risk report, the World Economic Forum (2023) indicated that extreme weather is among the most severe global threats. To address extreme weather events such as torrential heavy rains and heat waves, further advancements in weather forecasting are essential. There are two essential components for accurate weather forecasting: (1) numerical weather prediction (NWP) models that forecast future weather based on initial conditions, and (2) data assimilation, which integrates atmospheric observation data to estimate initial conditions for subsequent forecasts by NWP models.

Since NVIDIA issued the first artificial intelligence (AI) weather prediction model competitive to dynamical NWP models, FourCastNet, in February 2022 (Pathak et al. 2022, Bonev et al. 2023), deep learning-based weather prediction research has shown rapid growth. A number of AI weather prediction models have been proposed mainly by private information and technology (IT) companies such as GraphCast by Google DeepMind (Lam et al. 2023), Pangu-Weather by Huawei (Bi et al. 2023), ClimaX and Stormer by Microsoft (Nguyen et al. 2023), and Aurora by Microsoft (Bodnar et al. 2024). These machine learning approaches have been shown to be competitive with state-of-the-art NWP models (e.g., Kochkov et al. 2024). Progresses in AI-based weather prediction has been supported by the expansion of benchmark data and evaluation algorithms, such as WeatherBench (Rasp et al. 2020, 2024). Notably, most AI-based weather prediction models, including Pang-Weather, ClimaX, Stormer, and FourCastNet, use the Vision Transformer (ViT) neural network architecture (Vaswani et al. 2017, Dosovitski et al. 2020). The ViT, which has been explored in language models and image classifications, was demonstrated to be effective in weather prediction as well.

However, research that couples AI-based weather prediction models with data assimilation remains limited. This limitation is partially due to the fact that long-term sequential data assimilation experiments are needed for the evaluation of data assimilation systems, in contrast to weather prediction tasks that allow for parallel learning using benchmark data. Conventional data assimilation methods used in NWP systems can be categorized into three groups: variational methods, ensemble Kalman filters, and particle filters. There are strong mathematical similarities between neural networks and variational data assimilation, both of which minimize their cost functions using their differentiable models. Because auto-differentiation codes are always available for neural-network-based AI models, AI weather prediction models are considered compatible with variational data assimilation methods as in Xiao et al. (2023) and Adrian et al. (2024). On the other hand, recent studies have started to solve the inverse problem inherent in data assimilation by deep neural networks (McCabe and Brown 2021, Chen et al. 2023, Boucquet et al. 2024, Luk et al. 2024, Vaughan et al. 2024). There have been some studies employing ensemble Kalman filters for data-driven models (Hamilton et al. 2016, Penny et al. 2022, Chattopadhyay et al. 2022, 2023). However, no study has succeeded in employing ensemble Kalman filtering with global AI models of the atmosphere. Since AI models require significantly lower computational costs compared to dynamical NWP models, AI models offer benefits for ensemble-based methods, such as ensemble Kalman filters (EnKFs) and particle filters. Ensemble data assimilation at the global scale also allows us for assessing the capability of data assimilation with AI models to handle spatially

inhomogeneous observation networks and to maintain physically consistent multivariate error covariance across the entire atmosphere.

This study proposes using ensemble data assimilation for diagnosing AI-based weather prediction models. For that purpose, this study marks the first successful implementation of ensemble Kalman filter experiments with an AI weather prediction model to the best of the authors knowledge. We applied the ViT-based ClimaX (Nguyen et al. 2023) to data assimilation experiments using the available source code and experimental environments with necessary modifications. For data assimilation, we applied the local ensemble transform Kalman filter (LETKF) (Hunt et al. 2007), which is among the most widely used data assimilation methods in operational NWP centers such as the European Centre for Medium-Range Weather Forecasts (ECMWF), Deutscher Wetterdienst (DWD) and Japan Meteorological Agency (JMA). Using the coupled ClimaX-LETKF data assimilation system, we investigated several key aspects of AI-based weather prediction model, including whether the data assimilation cycles stably for the ClimaX AI weather prediction model using ensemble Kalman filters; whether AIbased ensemble weather prediction accurately represents flow-dependent background error variance and covariance. We also investigated whether techniques such as covariance inflation and localization, which are conventionally used in EnKFs for dynamical NWP models, are effective for AI weather prediction models. By addressing these research questions, we aim to advance the integration of AI weather prediction models with data assimilation techniques, toward the development of more accurate weather forecasting. While this study primarily aims to use ensemble data assimilation for diagnosing AI-based weather prediction models, our research also represents an important step toward enabling real-time update of the AI weather models with meteorological observations.

The rest of paper is organized as follows: section 2 describes the methods and experiments and section 3 presents the results. Finally, section 4 provides discussion and summary.

# 2 Methods and experiments

#### 2.1 ClimaX Model

60

61

6263

64

65 66

67

68

69 70

71

72

73

74

75

76

7778

79

80

81

82

83

8485

8687

88 89

90 91 The ClimaX (Nguyen et al. 2023) is a ViT-based AI weather prediction model for the global atmosphere. Variable tokenization and variable aggregation are the key components of the ClimaX architecture upon ViT, as they provide flexibility and generality. This study used the low-resolution version of ClimaX (version 0.3.1), with 64 and 32 zonal and meridional grid points, respectively, corresponding to a spatial resolution of  $5.625^{\circ} \times 5.625^{\circ}$ . The vertical model level was set at seven (900, 850, 700, 600, 500, 250 and 50 hPa).

By default, ClimaX is set to be trained against only five variables: geopotential at 500 hPa, temperature at 850 hPa, temperature at 2 m, zonal wind at 10 m, and meridional wind at 10 m. We updated ClimaX for data assimilation, which allowed the AI model to produce variables required for subsequent forecasts (Table 1). The updated ClimaX has state vectors including zonal wind, meridional wind, temperature, specific humidity, and geopotential at seven vertical layers along with three surface variables: 10-m zonal wind, 10-m meridional wind, and 2-m temperature. We also diagnosed surface pressure, which is a

required input for data assimilation, based on geopotential and surface elevation. Figure 1 shows the training curves of the default and updated ClimaX models verified against WeatherBench data (Rasp et al. 2020). Data for the period 2006–2015 were used for training, and data for 2016 were used for validation. Here we re-trained the ClimaX entirely with the additional outputs (i.e., no transfer learning). It took approximately 4 hours with four GPU of NVIDIA RTX 6000Ada. Anomaly correlation coefficients increased and root mean square errors (RMSEs) decreased in Figure 1, indicating successful training of the updated ClimaX model. Because more variables were predicted by the updated ClimaX than by the default ClimaX, more training steps were required.

#### 2.2 Local Ensemble Transform Kalman Filter (LETKF)

The LETKF is among the most widely used data assimilation methods in operational NWP centers such as ECMWF, DWD and JMA. The LETKF simultaneously computes analysis equations at every model grid point with the assimilation of surrounding observations within the localization cut-off radius. The ClimaX–LETKF system was developed based on the SPEEDY–LETKF system (Kotsuki et al. 2022) by replacing the SPEEDY weather prediction model with ClimaX. Our future research can readily be expanded to particle filter experiments because the Kotsuki et al. (2022) system includes local particle filters in addition to the LETKF.

Let  $\mathbf{X}_t \equiv \left\{\mathbf{x}_t^{(1)}, \dots, \mathbf{x}_t^{(m)}\right\}$  be an ensemble state matrix, whose ensemble mean and perturbation is given by  $\bar{\mathbf{x}}_t \ (\in \mathbb{R}^n)$  and  $\delta \mathbf{X}_t \equiv \left\{\mathbf{x}_t^{(1)} - \bar{\mathbf{x}}_t, \dots, \mathbf{x}_t^{(m)} - \bar{\mathbf{x}}_t\right\} \ (\in \mathbb{R}^{n \times m})$ , respectively. Here, n and m are the system and ensemble sizes. The superscript (i) and subscript t denote the ith ensemble member and indicates the time, respectively. The EnKFs, including LETKF, estimate error covariance  $\mathbf{P} \ (\in \mathbb{R}^{n \times n})$  according to sample estimates based on ensemble perturbation:

$$\mathbf{P} \approx \frac{1}{m-1} \delta \mathbf{X} \delta \mathbf{X}^T. \tag{1}$$

111 The analysis update equation of the LETKF is given by:

112 
$$\mathbf{X}_t^a = \bar{\mathbf{x}}_t^b \cdot \mathbf{1} + \delta \mathbf{X}_t^b \widetilde{\mathbf{P}}_t^a (\mathbf{Y}_t^b)^T \mathbf{R}_t^{-1} \left( \mathbf{y}_t^o - \overline{H_t(\mathbf{X}_t^b)} \right) \cdot \mathbf{1} + \left[ (m-1) \widetilde{\mathbf{P}}_t^a \right]^{1/2}, \tag{2}$$

113 
$$\widetilde{\mathbf{P}}_t^a = \left[ \frac{(m-1)}{\beta} \mathbf{I} + (\mathbf{Y}_t^b)^T \mathbf{R}_t^{-1} \mathbf{Y}_t^b \right]^{-1}, \tag{3}$$

where,  $\widetilde{\mathbf{P}}$  is the error covariance matrix in the ensemble space  $(\in \mathbb{R}^{m \times m})$ ,  $\mathbf{Y} \equiv \mathbf{H} \delta \mathbf{X}$  is the ensemble perturbation matrix in the observation space  $(\in \mathbb{R}^{p \times m})$ ,  $\mathbf{R}$  is the observation error covariance matrix  $(\in \mathbb{R}^{p \times p})$ ,  $\mathbf{y}$  is the observation vector  $(\in \mathbb{R}^p)$ , H is the observation operator that may be nonlinear,  $\mathbf{H}$   $(\in \mathbb{R}^{p \times n})$  is the Jacobian of linear observation operator matrix, and  $\mathbf{1}$  is a row vector whose all elements are 1  $(\in \mathbb{R}^m)$ . Here, p is the number of observations. The superscripts o, b, and a denote the observation, background, and analysis, respectively. The scalar  $\beta$  is a multiplicative inflation factor which inflates the background error covariance such that  $\mathbf{P}_t^b \to (1+\beta)\mathbf{P}_t^b$ . This study uses the Miyoshi (2011)'s approach, which estimates spatially varying inflation factors adaptively based on observation-space statistics (Desroziers et al. 2005).

Localization is a practically important technique for EnKFs to eliminate long-range erroneous correlations due to the sample estimates of **P** with a limited ensemble size (Houtekamer and Zhang, 2016). Although a larger localization can spread observation data information for grid points distant from observations, a larger localization scale can yield suboptimal error covariance because of sampling errors. The LETKF inflates the observation error variance to realizes the localization (Hunt et al. 2007) whose function is given by:

$$l = \begin{cases} \exp\left[-\frac{1}{2}\{(d_h/L_h)^2 + (d_v/L_v)^2\}\right] & if \ d_h 

## Figures

**Figure 1:** Training curves for the default and updated ClimaX models (dashed blue and solid orange lines) verified against WeatherBench data in 2016, as a function of the number of training steps. Each training step includes 64 training data in a mini batch. Panels (a-c) and (d-f) show anomaly correlation coefficients (ACCs) and root mean square errors (RMSEs). (a, d), (b, e) and (c, f) are geopotential at 500 hPa (m²/s²), temperature at 850 hPa and zonal wind at 500 hPa. There are no blue dashed lines in panels (c) and (f) because the default ClimaX model does not predict zonal wind at 500 hPa.

**Figure 2.** The observing network. Small black dots and red crosses represent model grid points and observing points, respectively.

# Weather Bench : (5th level T [K]; 500 hPa) (a): 2017010300 UTC (b): 2017020100 UTC (c): 2017050100 UTC

ClimaX Forecast : (5th level T [K]; 500 hPa)

Initialized at 2017010100

**Figure 3:** Spatial patterns of temperature (K) at 5th model level (500 hPa). Panels (a-c) are WeatherBench data. Panels (d-f) are forecasts by ClimaX initialized at 0000 UTC of January 1, 2017. Panels (a, d) show 0000 UTC of January 3, 2017, (b, e) show 0000 UTC of February 1, 2017, and (c, f) show 0000 UTC of May 1, 2017, respectively.

393394

395

396

**Figure 4:** Time series of global-mean root mean square errors (RMSEs) verified against WeatherBench data, and ensemble spreads for (a) temperature (K) and geopotential height (m) at the fifth model level (= 500 hPa). Thin and bold solid lines indicate 6-hourly RMSEs and their 7-day running means, respectively. Dashed lines indicate ensemble spreads. Black, purple, blue, green, and red lines indicate the ClimaX-LETKF experiments, at localization scales of  $L_h$ = 400, 500, 600, 700 and 800 km. The abscissa indicates the date (month/day) in 2017.

**Figure 5:** Global mean root mean square errors (RMSEs) verified against WeatherBench (WB) for (a) zonal wind (m/s), (b) meridional wind (m/s), (c) temperature (K), (d) specific humidity (g/kg), (e) geopotential height (m), and (f) surface pressure (hPa), as a function of the horizontal localization scales (km) averaged over July–December 2017. Colored bars and black diamonds indicate analysis (AN) and first-guess (FG) RMSEs, respectively. Blue, green, red, and purple bars in (a-e) represent 2nd, 3rd, 5th and 6th model levels (850, 700, 500, and 250 hPa, respectively). Gray bars in (f) represent surface pressure. The RMSEs of specific humidity at the 6th model level in (d) were too low to be shown.

**Figure 6:** Spatial patterns of difference between analysis (AN) and first-guess (FG) mean absolute errors (MAEs) for (a) zonal wind (m/s) at 850 hPa, (b) temperature (K) at 700 hPa, (c) geopotential height (m) at 500 hPa, and surface pressure (hPa), averaged over July–December 2017. Warm and cold colors represent improvements and degradations due to data assimilation. Results are for a localization scale of  $L_h = 500$  km. Black crosses indicate observing stations.

Figure 7: (a) Spatial pattern of the multiplicative inflation factor at the end of experiment on 1800UTC of December 31, 2017.
 (b) Time series of globally averaged inflation factors. Results are for a localization scale of L<sub>h</sub> = 600 km.