# Peer review of "Ensemble data assimilation to diagnose AI-based weather prediction"

_EGUsphere, 2024_

## Author Response (AR1)

**[Response to Reviewer's Comments]**

We are very grateful to the reviewers for their careful reviews and kindly giving us valuable and constructive comments and suggestions that we have generally accepted. Here, we provide our point-by-point responses whose P and L correspond to page and line numbers of the supplemental PDF file with trach changes.

\_\_\_\_\_\_

**[Reviewer 1: General Comments]**

The authors apply an ensemble Kalman filter (EnKF) to a machine learning (ML)-based model for weather prediction. This is an interesting and important study, which I believe deserves to be published after some revisions. In particular, the literature review is insufficient, and the paper is lacking some details.

Response: Thank you for the comment. We revised the manuscript following comments.

**[Reviewer 1: Issues]**

(1) "On the other hand, recent studies have started to solve the inverse problem inherent in data assimilation by deep neural networks". Some other papers can also be cited along these lines: McCabe & Brown (2021), Luk et al. (2024), and Bocquet et al. (2024).

Response: Thank you for sharing information. We added them in the revised manuscript (P2L53).

(2) "However, no study has succeeded in employing ensemble Kalman filtering with AI models." This statement should be made more specific: this is perhaps the first application of an EnKF to a global ML model of the atmosphere. However, there is older work applying EnKFs to ML models which should also be mentioned: Hamilton et al. (2016), Penny et al. (2022), and Chattopadhyay et al. (2022, 2023).

Response: Thank you for clarifications. We added them in the revised manuscript (P2L54).

(3) The authors say that the ClimaX model was retrained to predict additional variables. More detail would be helpful here. Was the entire training process repeated with the additional outputs, or did only some new part of the network have to be trained? Given the fact that it has been mostly tech companies training these ML models partly due to the computational limitations in academia, what were the computational resources involved in the training?

Response: We added description on this point (P4L94).

(4) "at larger localization scales ( $L_h = 700$  and 800 km), analysis RMSEs tended to be higher than the first guess RMSEs". I am somewhat confused by this claim. For  $L_h = 800$  km this makes sense, since the filter appears to diverge. But for  $L_h = 700$  km, if the filter is stable, then is it not contradictory for the

analysis RMSEs to be consistently higher than the first guess ones (since the error would just keep growing)? Of course, it can be the case that the RMSEs of some variables worsen with analysis, but I believe that there should be an overall reduction for the filter to be stable (unless the system is not chaotic).

Response: Thanks. We added discussion on this point (P6L176).

(5) "Our optimal localization scale for the 20-member ClimaX-LETKF was 600 km, which is significantly shorter than the 900-km scale of the 20-member LETKF experiment coupled with a dynamical NWP model". In Miyoshi and Kondo (2013), they find an optimal radius of 700 km in a similar setup, which seems not too different than what the authors get here with ClimaX. Unless there are systematic comparisons done it seems difficult to conclude that the optimal radius is significantly lower than for a dynamical model, and draw the resulting conclusions.

Response: Thank you for the comment. We agree that the difference is not necessarily significant, and have updated the text to reflect this point more cautiously (P7L213).

(6) "Another important property is that the ClimaX is less chaotic than dynamical NWP models, as indicated by the estimated inflation factor  $\beta$  diagnosed by observation-space statistics": I do not think this is a justified conclusion. It could be that ClimaX has higher model error, which is compensated for using the inflation.

Response: Thanks for your comment, we agree that the model error is another source of the stronger inflation. We added this point in the revised manuscript (P7L200).

(7) "It should be noted that we were unable to conduct observation system simulation experiments (k.a. OSSEs), which requires a natural run by ClimaX." I think that this paragraph is quite important and that it could go earlier in the paper (along with Figure 7). As it stands it is somewhat hidden in the discussion section.

Response: Revised as suggested (P5L147).

(8) "In other words, this suggests that ClimaX is unable to return to a meteorologically plausible attractor (or trajectory) while data assimilation enables the ClimaX to synchronize with the real atmosphere." This fact is actually proved rigorously in a simpler setting in Theorem 1 in Adrian et al. (2024): "our theory rigorously shows that if we have long-term filter accuracy with the true dynamics model F and a surrogate model F\_s that provides accurate short-term forecasts, we can achieve long-term filter accuracy with the surrogate dynamics." This should be mentioned.

Response: Thank you very much for sharing interesting property. We added this point (P6L155).

(9) In the discussion section, the authors discuss two major improvements in ML forecast models that would improve ensemble data assimilation: accurate covariances, and accurate error growth rates. In this

connection, it would be helpful to mention that a major area of research is in models that are trained to produce statistically accurate ensembles by using generative models (Price et al., 2024) or by training on probabilistic cost functions (Kochkov et al., 2024). Also, there has previously been research on enforcing the error growth rate during training as measured by the Lyapunov exponent (Platt et al., 2023).

Response: Thanks, we added descriptions on this point (P8L227).

(10) The quality of Figure 3 is quite low.

Response: The low resolution of the figure (Figure 4 in the revised manuscript) may have resulted from the automatic PDF generation process in the submission system. For reference, we have attached the original high-resolution version of the figure to this response.

**[Reviewer 1: Minor Issues]**

(1) "Since Google DeepMind issued the first artificial intelligence (AI) weather prediction model, GraphCast". I believe that FourCastNet was earlier in 2022, which seems to be the case based on the arXiv dates. In any case, there are many older attempts to forecast weather based on statistical/machine learning techniques. It would be more accurate to say that FourCastNet was the first ML weather prediction model competitive in skill with dynamical forecast models.

Response: Thank you for your clarification. Revised (P2L35).

(2) "A number of AI weather prediction models have been proposed even by private information and technology (IT) companies". In fact, all the leading models I am aware of have come from private companies, with the exception of the ECMWF's AI Forecasting System (AIFS).

Response: Revised (P2L36)

(3) "Nguyen" is misspelled as "Nguen" several times in the paper. Also, the authors cite "Nguen et al. (2023, 2024)" but there is no 2024 paper in the bibliography, just two 2023 papers. The second one has a 2024 version on arXiv, so perhaps this is what the authors meant to cite.

Response: Revised (P2L38).

(4) "grids" should be replaced with "grid points" throughout the manuscript.

Response: Revised throughout the manuscript.

(5) "Since AI models require significantly lower computational costs compared to dynamical NWP models, ensemble-based methods, such as ensemble Kalman filters (EnKFs) and particle filters, also offer benefits for AI models." I think, given this justification, that it would be more accurate to say that the AI models offer benefits for ensemble-based methods.

Response: Revised (P2L57).

(6) "We employed a series of data assimilation experiments over a year of 2017, which is not used for training and validation of the ClimaX. The ensemble size is 20, and their initial conditions were taken from WeatherBench data in 2006." Are you saying that the initial ensemble (for January 1, 2017) was taken from 2006 data? Did this come from consecutive days in 2006?

Response: Revised (P5L145).

(7) Figure 3 caption, typo "WeatharBench".

Response: Revised (The caption of Figure 4).

(8) "the experiment with  $L_h$  = 400 km kept reducing the RMSEs over a year, indicating that a too small localization scale is suboptimal". It would be good to clarify that the RMSEs kept reducing but are still higher than those of the other experiments, with the exception of  $L_h$  = 800 km.

Response: Revised (P6L167).

(9) "Negative and positive values indicate improvements and degradations due to data assimilation": needs a "respectively" at the end.

Response: Revised (P7L188).

(10) "Compared to our study, Kotsuki et al. (2017) estimated much smaller inflation factor for a global ensemble data assimilation system using a dynamical model": Please state the average inflation factors in Kotsuki et al. (2017) compared to yours.

Response: Revised (P7L204).

(11) "The estimated inflation factor of Kotsuki et al. (2017) was substantially smaller than our study." This sentence is essentially repeating the previous one.

Response: You are right. Corresponding sentence was removed in the revised manuscript (P7L203).

(12) "In addition, the ensemble-based error covariance was reasonable in sparsely observed regions, even according to AI weather prediction models." I'm not sure what is meant by "even according to AI weather prediction models".

Response: Revised (P8L240).

(13) In Figure 4 the acronym "WB" (I suppose meaning WeatherBench) is not defined.

Response: Revised (The caption of Figure 5).

\_\_\_\_\_

**[Reviewer 2 (Dr. Sibo Cheng): General Comments]**

This study explores the use of ensemble DA, in particular, LETKF to diagnose and improve AI-based weather prediction models. The authors use Microsoft's ClimaX v0.3.1 as a test case. Overall, I found the research idea interesting and potentially impactful for the field. However, I believe the presentation of the methodology could be improved to highlight the contribution of the author's work and make the paper easier to follow.

Response: Thank you for the comment. We revised the manuscript following comments.

**[Reviewer 2 (Dr. Sibo Cheng): Comments]**

(1) Introduction: Many recent studies have combined data assimilation with Al-based weather prediction models, although most rely on variational data assimilation as mentioned by the authors. The authors should further summarize the existing literature and clearly highlight their contribution in relation to these existing approaches. In general, many efforts have also been given to combine DA (both variational and Kalman-type) with machine learning predictive (or surrogate) models (not necessary for weather prediction). It would be beneficial to explore how the methods developed in this paper could be applied to other fields where Al predictive models and real-time observations are available.

Response: Thank you for this helpful comment. We have added additional references to recent studies combining data assimilation with Al-based models, including both variational and Kalman-type approaches. These additions clarify the positioning of our work relative to existing literature. We also included a brief discussion on how the proposed method could be extended to other domains where Al predictive models and real-time observations are available (P2L58, P9L256).

- (2) Introduction: It seems the main objective of using LETKF in this paper is to diagnose Al-based models but can it be used to make real-time update of the Al weather model based on real-time observations? to demonstrate that, one would probably need numerical experiments with higher resolution data?

  Response: Thanks for your comments. We added descriptions in Introduction and Summary (P2L75, P9L249)
- (3) Methodology: in Section 2.2, the authors are encouraged to define the state and observation variables, as well as the observation operator, in the context of the predictive climate model X used in this paper. It seems that these important information is only available in table 1?

  Response: We added descriptions (P3L89, P5L141)

(4) Methodology: How sensitive are the results to localization radius, inflation methods, or ensemble size? Could variational DA (with the help of autodifferentiation codes as mentioned by the authors) methods

**with AI weather models outperform LETKF?**

Response: Thank you for this valuable comment. In this study, we used fixed settings for inflation method, and ensemble size, and did not perform a systematic sensitivity analysis. We agree that evaluating the sensitivity of the system to these parameters is important, and we consider this a key direction for future work. While our work focuses on ensemble-based DA, we agree that comparing with variational approaches would be an important topic for future investigation. However, we believe that detailed comparisons and sensitivity experiments should be conducted after addressing the two major challenges identified in this study—namely, the lack of physically consistent multivariate error covariances and the insufficient representation of error growth in Al-based ensemble forecasts. We added discussion on this point (P8L245).

- (5) Results: How does the relatively coarse resolution (5.625°) affect the diagnosis of localized weather phenomena? Would ClimaX + LETKF behave differently if applied at higher spatial resolution? Response: We added discussion on this point (P8L232).
- (6) Could we use ensemble-based DA (such as the one developed in this paper) to generate synthetic training data for AI models in under-observed regions?

Response: Thanks for your insightful comments. We added discussion on this point (P9L251).